# Mutual exclusivity as a challenge for deep neural networks

## Abstract

Strong inductive biases allow children to learn in fast and adaptable ways. Children use the mutual exclusivity (ME) bias to help disambiguate how words map to referents, assuming that if an object has one label then it does not need another. In this paper, we investigate whether or not standard neural architectures have an ME bias, demonstrating that they lack this learning assumption. Moreover, we show that their inductive biases are poorly matched to lifelong learning formulations of classification and translation. We demonstrate that there is a compelling case for designing neural networks that reason by mutual exclusivity, which remains an open challenge.

## 1 Introduction

Children are remarkable learners, and thus their inductive biases should interest machine learning researchers. To help learn the meaning of new words efficiently, children use the "mutual exclusivity" (ME) bias – the assumption that once an object has one name, it does not need another (Markman & Wachtel, 1988) (Figure 1). In this paper, we examine whether or not standard neural networks demonstrate the mutual exclusivity bias, either as a built-in assumption or as a bias that develops through training. Moreover, we examine common benchmarks in machine translation and object recognition to determine whether or not a maximally efficient learner should use mutual exclusivity.

When children endeavour to learn a new word, they rely on inductive biases to narrow the space of possible meanings. Children learn an average of about 10 new words per day from the age of one until the end of high school (Bloom, 2000), a feat that requires managing a tractable set of candidate meanings. A typical word learning scenario has many sources of ambiguity and uncertainty, including ambiguity in the mapping between words and referents. Children hear multiple words and see multiple objects within a single scene, often without clear supervisory signals to indicate which word goes with which object (Smith & Yu, 2008).

Show me the "dax"

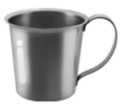 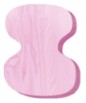

Figure 1: The mutual exclusivity task used in cognitive development research (Markman & Wachtel, 1988). Children tend to associate the novel word ("dax") with the novel object (right).

The mutual exclusivity assumption helps to resolve ambiguity in how words maps to their referents. Markman & Wachtel (1988) examined scenarios like Figure 1 that required children to determine the referent of a novel word. For instance, children who know the meaning of "cup" are presented with two objects, one which is familiar (a cup) and another which is novel (an unusual object). Given these two objects, children are asked to "Show me a dax," where "dax" is a novel nonsense word. Markman and Wachtel found that children tend to pick the novel object rather than the familiar object. Although it is possible that the word "dax" could be another word for referring to cups, children predict that the novel word refers to the novel object – demonstrating a "mutual exclusivity" bias that familiar objects do not need another name. This is only a preference; with enough evidence, children must eventually override this bias to learn hierarchical categories: a Dalmatian can be called a "Dalmatian," a "dog", or a "mammal" (Markman & Wachtel, 1988; Markman, 1989). As an often useful but sometimes misleading cue, the ME bias guides children when learning the words of their native language.

It is instructive to compare word learning in children and machines, since word learning is also a widely studied problem in machine learning and artificial intelligence. There has been substantial

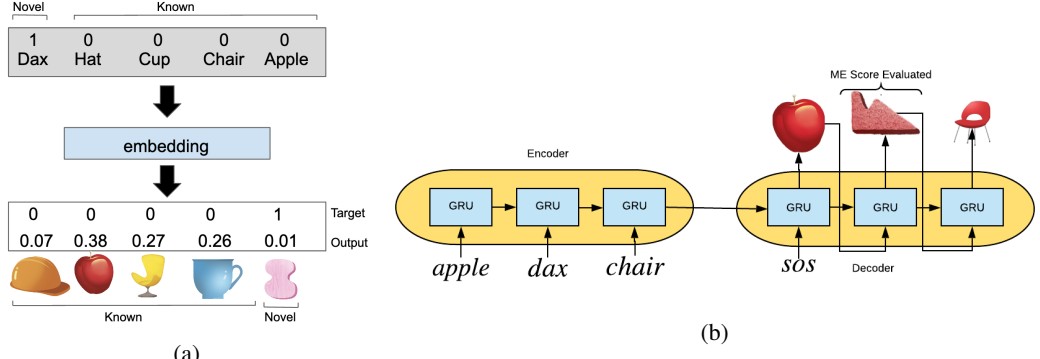

Figure 2: Evaluating mutual exclusivity in a feedforward (a) and seq2seq (b) neural network. (a) After training on a set of known objects, a novel label ("dax") is presented as a one-hot input vector. The network maps this vector to a one-hot output vector representing the predicted referent, through an intermediate embedding layer and an optional hidden layer (not shown). A representative output vector produced by a trained network is shown, placing almost all of the probability mass on known outputs. (b) A similar setup for mapping sequences of labels to their referents. During the test phase a novel label "dax" is presented and the ME Score at that output position is computed.

recent progress in object recognition, much of which is attributed to the success of deep neural networks and the availability of very large datasets (LeCun et al., 2015). But when only one or a few examples of a novel word are available, deep learning algorithms lack human-like sample efficiency and flexibility (Lake et al., 2017). Insights from cognitive science and cognitive development can help bridge this gap, and ME has been suggested as a psychologically-informed assumption relevant to machine learning (Lake et al., 2019). In this paper, we examine standard neural networks to understand if they have an ME bias. Moreover, we analyze whether or not ME is a good assumption in lifelong variants of common translation and object recognition tasks.

## 2 RELATED WORK

Children utilize a variety of inductive biases like mutual exclusivity when learning the meaning of words (Bloom, 2000). Previous work comparing children and neural networks has focused on the shape bias – an assumption that objects with the same name tend to have the same shape, as opposed to color or texture (Landau et al., 1988). Children acquire a shape bias over the course of language development (Smith et al., 2002), and neural networks can do so too, as shown in synthetic learning scenarios (Colunga & Smith, 2005; Feinman & Lake, 2018) and large-scale object recognition tasks (Ritter et al., 2017) (see also Id et al. (2018) and Brendel & Bethge (2019) for alternative findings). This bias is related to how quickly children learn the meaning of new words (Smith et al., 2002), and recent findings also show that guiding neural networks towards the shape bias improves their performance (Geirhos et al., 2019). In this work, we take initial steps towards a similar investigation of the ME bias in neural networks. Compared to the shape bias, ME has broader implications for machine learning systems; as we show in our analyses, the bias is relevant beyond object recognition.

Closer to the present research, Cohn-Gordon & Goodman (2019) analyzed an ME-like effect in neural machine translation systems at the sentence level, rather than the word level considered in developmental studies and our analyses here. Cohn-Gordon & Goodman (2019) showed that neural machine translation systems often learn many-to-one sentence mappings that result in meaning loss, such that two different sentences (meanings) in the source language are mapped to the same sentence (meaning) in the target language. Using a trained network, they show how a probabilistic pragmatics model (Frank & Goodman, 2012) can be used as a post-processor to preserve meaning and encourage one-to-one mappings. These sentence-level biases do not necessarily indicate how models behave at the word level, and we are interested in the role of ME during learning rather than as a post-processing step. Nevertheless, Cohn-Gordon and Goodman's results are important and encouraging, raising the possibility that ME could aid in training deep learning systems.

## 3 DO NEURAL NETWORKS REASON BY MUTUAL EXCLUSIVITY?

In this section, we investigate whether or not standard neural network models have a mutual exclusivity bias. Paralleling the developmental paradigm (Markman & Wachtel, 1988), ME is analyzed by presenting a novel stimulus ("Show me the dax") and asking models to predict which outputs (meanings) are most likely. The strength of the bias is operationalized as the aggregate probability mass placed on the novel rather than the familiar meanings.

Our analyses relate to classic experiments by Marcus on whether neural networks can generalize outside their training space (Marcus, 1998; 2003). Marcus showed that a feedforward autoencoder trained on arbitrary binary patterns fails to generalize to an output unit that was never activated during training. Our aim is to study whether standard architectures can recognize and learn a more abstract pattern – a perfect one-to-one mapping between input symbols and output symbols. Specifically, we are interested in model predictions regarding unseen meanings given a novel input. We also test for ME using modern neural networks in two settings using synthetic data: classification (feedforward classifiers) and translation (sequence-to-sequence models; as reported in Appendix A).

### 3.1 CLASSIFICATION

**Synthetic data.** We consider a simple one-to-one mapping task inspired by Markman & Wachtel (1988). Translating this into a synthetic experiment, input units denote words and output units denote objects. Thus, the dataset consists of 100 pairs of input and output patterns, each of which is a one-hot vector of length 100. Each input vector represents a label (e.g., 'hat', 'cup', 'dax') and each output vector represents a possible referent object (meaning). Figure 2a shows the input and output patterns for the 'dax' case, and similar patterns are defined for the other 99 input and output symbols. A one-to-one correspondence between each input symbol and each output symbol is generated through a random permutation, and there is no structure to the data beyond the arbitrary one-to-one relationship.

Models are trained on 90 name-referent pairs and evaluated on the remaining 10 test pairs. No model can be expected to know the correct meaning of each test name – there is no way to know from the training data – but several salient patterns are discoverable. First, there is a precise one-to-one relationship exemplified by the 90 training items; the 10 test items can be reasonably assumed to follow the same one-to-one pattern, especially if the network architecture has exactly 10 unused input symbols and 10 unused output symbols. Second, the perfect one-to-one relationship ensures a perfect ME bias in the structure of the data. Although the learner does not know precisely *which new output symbol* a new input symbol refers to, it should predict that the novel input symbol will correspond to *one of the novel output symbols*. An ideal learner should discover that an output unit with a known label does not need another – in other words, it should utilize ME to make predictions.

**Mutual exclusivity.** We ask the neural network to "Show me the dax" by activating the "dax" input unit and asking it to select amongst possible referents (similar to Figure 1). The network produces a probability distribution over candidate referents (see Figure 2a), and can make relative (two object) comparisons by isolating the two relevant scores. To quantify the overall propensity toward ME, we define an "ME score" that measures the aggregate probability assigned to all of the novel output symbols as opposed to familiar outputs, corresponding to better performance on the classic forced choice ME task. Let us denote the training symbol by $\mathcal{Y}$, drawn from the data distribution $(\mathcal{X}, \mathcal{Y}) \sim \mathcal{D}$ and the held out symbols $\mathcal{Y}'$ drawn from $(\mathcal{X}', \mathcal{Y}') \sim \mathcal{D}'$. The mutual exclusivity score is the sum probability assigned to unseen output symbols $\mathcal{Y}'$ when shown a novel input symbol $x \in \mathcal{X}'$

$$\text{ME Score} \ = \ \frac{1}{|\mathcal{D}'|} \sum_{(x_i, y_i) \in \mathcal{D}'} \sum_{y \in Y'} P(f_{net}(x) = y | x_i), \tag{1}$$

averaged over each of the test items. An ideal learner that has discovered the one-to-one relationship in the synthetic data should have a perfect ME score of 1.0. In Figure 2a, the probability assigned to the novel output symbol is 0.01 and thus the corresponding ME Score is 0.01. The challenge is to get a high ME score for novel (test) items while also correctly classifying known (training) items.

**Neural network architectures.** A wide range of standard neural networks are evaluated on the mutual exclusivity test. We use an embedding layer to map the input symbols to vectors of size 20

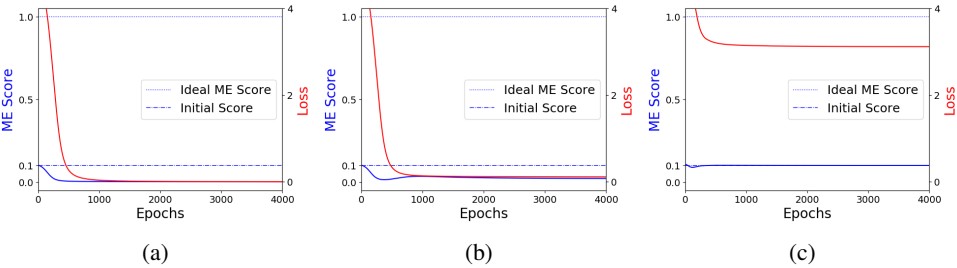

Figure 3: Evaluating mutual exclusivity on synthetic categorization tasks. ME Score (solid blue) and the cross-entropy loss (solid red) are plotted against the epochs of training. The configurations in the settings shown were: (a) Results for a model with an embedding, hidden, and classification layers, (b) Results for a model with embedding and classification layers trained with a weight decay factor of 0.001, and (c) Results for a model with an embedding and classification layer trained with an entropy regularizer.

or 100, followed optionally by a hidden layer, and then by a 100-way softmax output layer. The networks are trained with different activation functions (ReLUs (Nair & Hinton, 2010), TanH, Sigmoid), optimizers (Adam (Kingma & Ba, 2015), Momentum, SGD), learning rates (0.1, 0.01, 0.001) and regularizers (weight decay, batch-normalisation (Ioffe & Szegedy, 2015), dropout (Srivastava et al., 2014b), and entropy regularization (see Appendix B.1)). The models are trained to maximize log-likelihood. All together, we evaluated over 400 different models on the synthetic ME task.

**Results.** Several representative training runs with different architectures are shown in Figure 3. An ideal learner that has discovered the one-to-one pattern should have a mutual exclusivity of 1; for a novel input, the network would assign all the probability mass to the unseen output symbols. In contrast, none of the configurations and architectures tested behave in this way. As training progresses, the mutual exclusivity score (solid blue line; Figure 3) tends to fall along with the training loss (red line). In fact, almost all of the networks acquire a strong *anti-mutual exclusivity bias*, transitioning from a initial neutral bias to placing most or all of the probability mass on familiar outputs (seen in Figure 4). An exception to this pattern is the entropy regularized model, which maintains a score equivalent to an untrained network. In general, trained models strongly predict that a novel input symbol will correspond to a

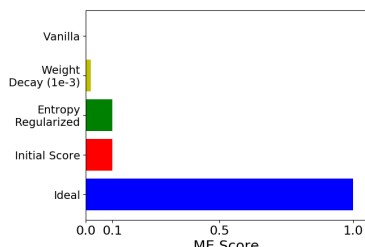

Figure 4: Ideal and untrained ME scores compared with the ME scores of a few learned models.

known rather than unknown output symbol, in contradiction to ME and the organizing structure of the synthetic data.

Further informal experiments suggest our results cannot be reduced to simply not enough data: these architectures do not learn this one-to-one regularity regardless of how many input/output symbols are provided in the training set. Even with thousands of training examples demonstrating a one-to-one pattern, the networks do not learn this abstract principle and fail to capture this defining pattern in the data. Other tweaks were tried in an attempt to induce ME, including eliminating the bias units or normalizing the weights, yet we were unable to find an architecture that reliably demonstrated the ME effect.

## 3.2 DISCUSSION

The results show that standard neural networks fail to reason by mutual exclusivity when trained in a variety of typical settings. The models fail to capture the perfect one-to-one mapping (ME bias) seen in the synthetic data, predicting that new symbols map to familiar outputs in a many-to-many fashion.

Although our focus is on neural networks, this characteristic is not unique to this model class. We posit it more generally affects flexible models trained to maximize log-likelihood. In a trained network, the optimal activation value for an unused output node is zero: for any given training example, increasing value of an unused output simply reduces the available probability mass for the

| Name | Languages | Sentence Pairs | Vocabulary Size |
|------|-----------|----------------|-----------------|
| IWSLT'14 (Freitag et al., 2014) | Eng.-Vietnamese | ~133K | 17K(en), 7K(vi) |
| WMT'14 (Luong et al., 2015) | Eng.-German | ~4.5M | 50K(en), 50K(de) |
| WMT'15 (Luong & Manning, 2016) | Eng.-Czech | ~15.8M | 50K(en), 50K(cs) |

Table 1: Datasets used to analyze ME in machine translation.

target output. Using other loss functions could result in different outcomes, but we also did not find that weight decay and entropy regularization of reasonable values could fundamentally alter the use of novel outputs. In the next section, we investigate if the lack of ME could hurt performance on common learning tasks such as machine translation and image classification.

## 4    SHOULD NEURAL NETWORKS REASON BY MUTUAL EXCLUSIVITY?

Mutual exclusivity has implications for a variety of common learning settings. Mutual exclusivity arises naturally in lifelong learning settings, which more realistically reflect the "open world" characteristics of human cognitive development. Unlike epoch-based learning, a lifelong learning agent does not assume a fixed set of concepts and categories. Instead, new concepts can be introduced at any point during learning. An intelligent learner should be sensitive to this possibility, and ME is one means of intelligently reasoning about the meaning of novel stimuli.

Children and adults learn in an open world with some probability of encountering a new class at any point, resembling the first epoch of training a neural net only. Moreover, the distribution of categories is neither uniformly distributed nor randomly shuffled (Smith & Slone, 2017). To simulate these characteristics, we construct lifelong learning scenarios using standard benchmarks as described below.

### 4.1    MACHINE TRANSLATION

In this section, we investigate if mutual exclusivity could be a helpful bias when training machine translation models in a lifelong learning paradigm. From the previous experiments, we know that the type of sequence-to-sequence (seq2seq) models used for translation acquire an anti-ME bias over the course of training (see Appendix A). Would a translation system benefit from assuming that a single word in the source sentence maps to a single word in the target sentence, and vice-versa? This assumption is not always correct since synonymy and polysemy are prevalent in natural languages, and thus the answer to whether or not ME holds is not absolute. Instead, we seek to measure the degree to which this bias holds in lifelong learning on real datasets, and compare this bias to the inductive biases of models trained on these datasets. The data for translation provides a somewhat natural distribution over the frequency at which different words are observed (there are words that appear much more frequently than the others). This allows us to use a single pass through the dataset as a proxy for lifelong translation learning.

**Datasets.** We analyze three common datasets for machine translation, each consisting of pairs of sentences in two languages (Table 1). The vocabularies are truncated based on word frequency in accordance with the standard practices for training neural machine translation models (Freitag et al., 2014; Luong et al., 2015; Luong & Manning, 2016).

**Mutual exclusivity.** There are several ways to operationalize mutual exclusivity in a machine translation setting. Mutual exclusivity could be interpreted as whether a new word in the source sentence ("Xylophone" in English) is likely to be translated to a new word in the target sentence ("Xylophon" in German), as opposed to a familiar word. Since the word alignments are difficult to determine and not provided with the datasets, we instead measure a reasonable proxy: if a new word is encountered in the source sequence, is a new word also encountered in the target sentence? For a source sentence $S$ and an arbitrary novel word $N_S$, and a target sentence $T$ and a novel word $N_T$, we measure a dataset's ME Score as the conditional probability $P(N_T \in T | N_S \in S)$. A hypothetical translation model could compute whether or not $N_S \in S$ by checking if the present word is absent from the vocabulary-so-far during the training process. Thus this conditional probability is an easily-computable cue for determining whether or not a model should expect a novel output word.

For the three datasets, we consider both forward and backward translation to get six scenarios for analysis. The probability $P(N_T \in T | N_S \in S)$ is estimated for a sample of 100 randomly shuffled sequences of the dataset sentence pairs. See Appendix B.3 for details on calculating the base rate $P(N_T \in T)$.

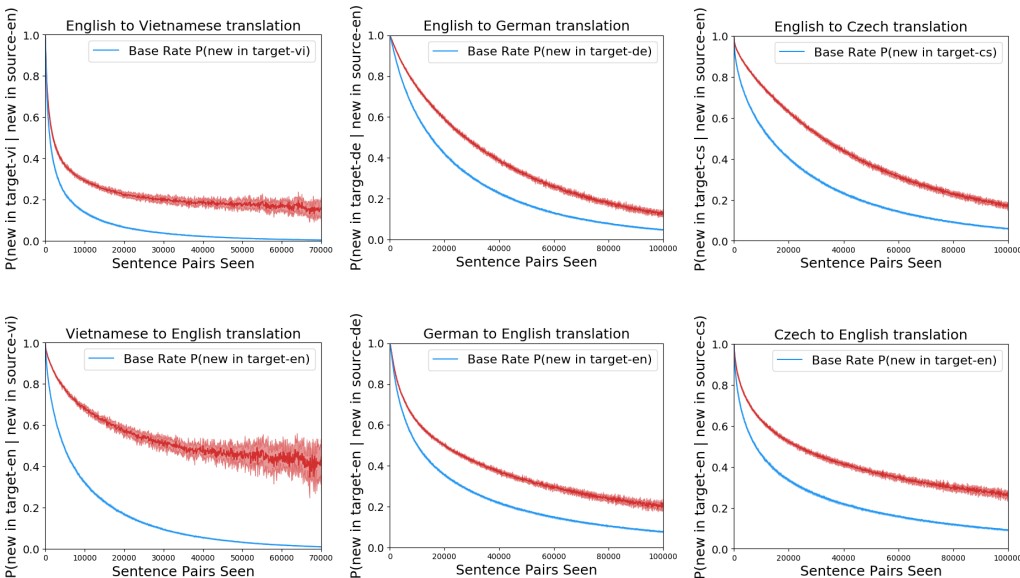

Figure 5: Analysis of mutual exclusivity in machine translation datasets. The plots show the conditional probability of encountering a new word in the target sentence, if a new word is present in the source sentence (y-axis; red line). Also plotted is the base rate of encountering a new target word (blue line). These quantities are measured as at different points during training (x-axis). Errors bars are standard deviations.

**Results and Discussion.** The measures of conditional probability in the six scenarios are shown in Table 2. There is a consistent pattern through the trajectory of early learning: the conditional probability $P(N_T \in T | N_S \in S)$ is high initially for thousands of initial sentence presentations, but then wanes as the network encounters more samples from

| Score | En-Vi | Vi-En | En-De | De-En | En-Cs | Cs-En |
|-------|-------|-------|-------|-------|-------|-------|
| 0.9 | 0.3K | 2K | 4K | 3K | 4K | 3K |
| 0.5 | 3K | 40K | 37K | 30K | 40K | 30K |
| 0.1 | 90K | 120K | 120K | 140K | 130 | 150K |

Table 2: Number of sentences after which the ME Score $P(N_T \in T | N_S \in S)$ falls below threshold.

the dataset. For a large part of the initial training, a seq2seq model would benefit from predicting that previously unseen words in the source language are more likely to map to unseen words in the target language. Moreover, this conditional probability is always higher than the base rate of encountering a new word, indicating that conditioning on the novelty of the input provides substantial additional signal for predicting novelty in the output. Nevertheless, even the base rate suggests that a model should expect novel words with some regularity in our settings. This is in stark contrast to the synthetic results showing that seq2seq models quickly acquire an anti-ME assumption (see Appendix A), and their expectation of mapping novel inputs to novel outputs decays rapidly as training progresses (Appendix Figure 7).

## 4.2 IMAGE CLASSIFICATION

Similar to translation, we examine if object classifiers would benefit from reasoning by mutual exclusivity during training processes that mirror lifelong learning. To study this, when selecting an image for training, we sample the class from a power law distribution (see Appendix B.2) such that the model is more likely to see certain classes (Smith & Slone, 2017).

Ideally, we would model the probability that an object belongs to a novel class based on its similarity to previous samples seen by the model (e.g., outlier detection). Identifying that an image belongs to a novel class is non-trivial, and instead we calculate the base rate for classifying an image as "new" while a learner progresses through the dataset. The set of classes not seen by the model are referred to as "new" here. This measure can be seen as a lower bound on the usefulness of ME through the standard training process, since this calculation assumes a blind learner that is unaware of any novelty signal present in the raw image.

**Datasets.** This section examines the Omniglot dataset (Lake et al., 2015) and the ImageNet dataset (Deng et al., 2009). The Omniglot dataset has been widely used to study few-shot learning, consisting of 1623 classes of handwritten characters with 20 images per class. The ImageNet dataset consists of about 1.2 million images from 1000 different classes.

**Mutual exclusivity.** To measure ME throughout the training process, we examine if an image encountered for the first time while training belongs to a class that has not been seen before. This is operationalized as the probability of encountering an image from a new class $N$ as a function of the number of images seen so far $t$, $P(N|t)$ (see Appendix B.5). This analysis is agnostic to the content of the image and whether or not it is a repeated item; it only matters whether or not the class is novel. As before, the analysis is performed using ten random runs through the dataset.

We contrast the statistics of the datasets by comparing them to the ME Score (Equation 1) of neural network classifiers trained on the datasets. See Appendix B.4 for details about the models and their training procedures. The probability mass assigned to the unseen classes by the network is recorded after each optimizer step, as computed using Equation 1.

**Results and Discussion.** The results are summarized in Figure 6 and Table 3. The probability that a new image belongs to an unseen class $P(N|t)$ is higher than the ME score of the classifier through most of the learning phase. Comparing the statistics of the datasets to the inductive biases in the classifiers, the ME score for the classifiers is substantially lower than the baseline ME measure in the dataset,

| Score | Omniglot | Omniglot Classifier | Imagenet | Imagenet Classifier |
|---|---|---|---|---|
| 0.2 | 24,304 | 2,144 | 1,280 | 2,048 |
| 0.1 | 99,248 | 22,912 | 8,448 | 3,072 |
| 0.05 | 160,608 | 43,328 | 111,872 | 8,960 |

Table 3: Number of images after which the ME Score falls below threshold.

$P(N|t)$ (Table 3). For instance, the ImageNet classifier drops its ME score below 0.05 after about 8,960 images, while the approximate ME measure for the dataset shows that new classes are encountered at above this rate until at least 111,000 images. We found that higher learning rates can force the probabilities assigned to unseen classes to zero on ImageNet after just a single gradient step.

These results suggest that neural classifiers, with their bias favoring frequent rather than infrequent outputs for novel stimuli, are not well-suited to lifelong learning challenges where such inferences are critical. Although we examined classifiers trained in an online fashion, we would expect similar results when we train them using replay or epoch-based training setups, where repeated presentation of past examples would only strengthen the anti-ME bias. These classifiers are hurt by their lack of ME and their failure to consider that new stimuli likely map to new classes. Ideally, a learning algorithm should be capable of leveraging the image content, combined with its own learning maturity, to decide how strongly it should reason by ME. Instead, standard models and training procedures do not provide these capabilities and do not utilize this important inductive bias observed in cognitive development.

## 5   GENERAL DISCUSSION

Children use the mutual exclusivity (ME) bias to learn the meaning of new words efficiently, yet standard neural networks learn very differently. Our results show that standard deep learning algorithms lack the ability to reason with ME, including feedforward networks and recurrent sequence-to-sequence models trained to maximize log-likelihood with common regularizers. Beyond simply lacking this bias, these networks learn an anti-ME bias, preferring to map novel inputs to familiar and frequent (rather than unfamiliar) output classes. Our results also show that these characteristics

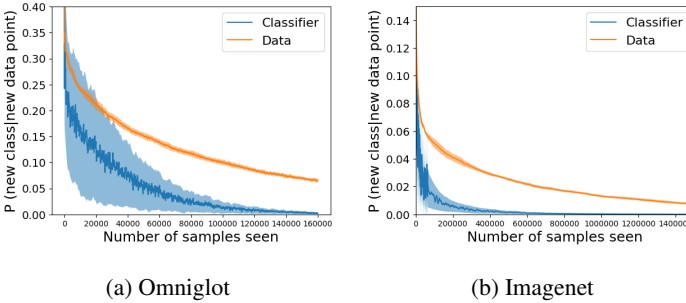

(a) Omniglot  (b) Imagenet

Figure 6: Analysis of mutual exclusivity in classification datasets. The plots show the probability that a new input image belongs to an unseen class $P(N|t)$, as a function of the number of images $t$ seen so far during training (blue), with its standard deviation. This measure is contrasted with the ME score of a neural network classifier trained through a similar run of the dataset (orange).

are poorly matched to more realistic lifelong learning scenarios where novel classes can appear at any point, as demonstrated in the translation and classification experiments presented here. Neural nets may be currently stymied by their lack of ME bias, ignoring a powerful assumption about the structure of learning tasks.

Mutual exclusivity is relevant elsewhere in machine learning. Recent work has contrasted the ability of humans and neural networks to learn compositional instructions from just one or a few examples, finding that neural networks lack the ability to generalize systematically (Lake & Baroni, 2018; Lake et al., 2019). The authors suggest that people rely on ME in these learning situations (Lake et al., 2019), and thus few-shot learning approaches could be improved by utilizing this bias as well.

In our analyses, we show that neural networks tend to learn the opposite bias, preferring to map novel inputs to familiar outputs. More generally, ME can be generalized from applying to "novel versus familiar" stimuli to instead handling "rare versus frequent" stimuli (e.g., in translation, rare source words may map to rare target words). The utility of reasoning by ME could be extended to early stages of epoch based learning too. For example, during epoch-based learning, neural networks take longer to acquire rare stimuli and patterns of exceptions (McClelland & Rogers, 2003), often mishandling these items for many epochs by mapping them to familiar responses. Another direction for future work is studying how the ME bias should interact with hierarchical categorization tasks. We posit that the ME assumption will be increasingly important as learners tackle more continual, lifelong, and large-scale learning challenges (Mitchell et al., 2018).

Mutual exclusivity is an open challenge for deep neural networks, but there are promising avenues for progress. The ME bias will not be helpful for every problem, but it is equally clear that the status quo is sub-optimal: models should not have a strong anti-ME bias regardless of the task and dataset demands. Ideally, a model would decide autonomously how strongly to use ME (or not) based on the demands of the task. For instance, in our synthetic example, an ideal learner would discover the one-to-one correspondence and use this perfect ME bias as a meta-strategy. If the dataset has more many-to-one correspondences, it would adopt another meta-strategy. This meta-strategy could even change depending on the stage of learning, yet such an approach is not currently available for training models. Previous cognitive models of word learning have found ways to incorporate the ME bias (Kachergis et al., 2012; McMurray et al., 2012; Frank et al., 2009; Lambert et al., 2005), although in ways that do not generalize to training deep neural networks. While successful in some domains, these models are highly simplified or require built-in mechanisms for implementing ME, making them so far impractical for use in realistic settings. As outlined above, it would be ideal to acquire a ME bias via meta learning or learning to learn (Allen et al., 2019; Snell et al., 2017), with the advantage of calibrating the bias to the dataset itself rather than assuming its strength a priori. For example, the meta learning model of Santoro et al. (2016) seems capable of learning an ME bias, although it was not specifically probed in this way. Recent work by Lake (2019) demonstrated that neural nets can learn to reason by ME if trained explicitly to do so, showing these abilities are within the repertoire of modern tools. However acquiring ME is just one step toward the goal proposed here: using ME to facilitate efficient lifelong learning or large-scale classification and translation.

In conclusion, standard deep neural networks do not naturally reason by mutual exclusivity, but designing them to do so could lead to faster and more flexible learners. There is a compelling case for building models that learn through mutual exclusivity.

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

APPENDIX

## A DO SEQUENCE-TO-SEQUENCE MODELS REASON BY MUTUAL EXCLUSIVITY?

**Synthetic data.** We evaluate if a different class of models, sequence-to-sequence (seq2seq) neural networks (Sutskever et al., 2014), take better advantage of ME structure in the data. This popular class of models is used in machine translation and other natural language processing tasks, and thus the nature of their inductive biases is critical for many applications. As in the previous section, we create a synthetic dataset that has a perfect ME bias: each symbol in the source maps to exactly one symbol in the target. We also have a perfect alignment in the dataset, so that each token in the source corresponds to the token at the same position in the target. The task is illustrated in Figure 2b. We consider translation from sequences of words to sequences of referent symbols.

The dataset consists of 20 label-referent pairings. Ten pairs are used to train the model and familiarize the learning algorithm with the task. The remaining ten pairs were used in the test phase. To train the model, we generate 1000 sequences of words whose lengths range from 1 to 5. To generate sequences for the test phase, words in the training sequences are replaced with new words with a probability of 0.2. Thus, 1000 sequences are used to test for ME. The ME score is evaluated using Equation 1 at positions in the output sequence where the corresponding input is new. As shown in Figure 2b, the ME score is evaluated in the second position which corresponds to the novel word "dax," using the probability mass assigned to the unseen output symbols.

**Neural network architectures.** We probe a seq2seq model that has a recurrent encoder using Gated Recurrent Units (GRUs) (Cho et al., 2014) and a GRU decoder. Both the encoder and decoder had embedding and hidden sizes of 256. Dropout (Srivastava et al., 2014a) with a probability of 0.5 was used during training. The networks are trained using an Adam (Kingma & Ba, 2015) optimizer with a learning rate of 0.001 and a log-likelihood loss. Two versions of the network were trained with and without attention (Luong et al., 2015).

**Results.** Results are shown in Figure 7 and confirm our findings from the feedforward network. The ME score falls to zero within a few training iterations, and the networks fail to assign any substantial probability to the unseen classes. The networks achieve a perfect score on the training set, but cannot extrapolate the one-to-one mappings to unseen symbols. Again, not only do seq2seq models fail to show a mutual exclusivity bias, they acquire a *anti-mutual exclusivity bias* that goes against the structure of the dataset.

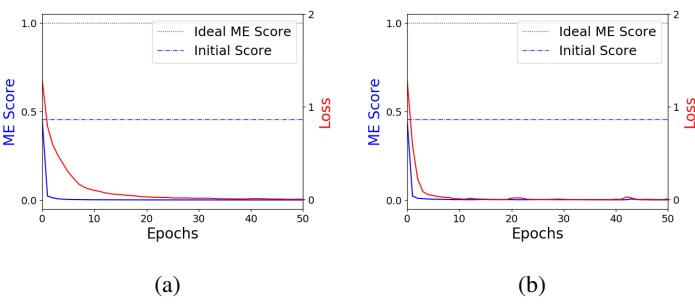

(a)  (b)

Figure 7: Results for the synthetic seq2seq task. The configurations shown in the setting are (a) Results for a seq2seq GRU without attention (b) Results for a seq2seq GRU with attention.

## B ADDITIONAL DETAILS

### B.1 ENTROPY REGULARIZER

The entropy regularizer is operationalized by subtracting the entropy of prediction from the loss. Thus, the model is penalized for being overly confident about its prediction. We found that the entropy regularizer produces an ME score that stays constant across training, at the cost of the model being less confident about predictions made for seen classes.

### B.2 SAMPLING USING A POWER LAW TO SIMULATE LIFELONG LEARNING

To simulate a naturalistic lifelong learning scenario, we try to sample a few classes more frequently than the others. To do this, we sample the classes for training using a power law distribution. Weights

are assigned to the classes using the following formula,

$$W(c) = \frac{1}{c^{1.5}}$$

where $c$ is the index of the class. After the class for training is chosen, we uniformly sample from the images of that class for training.

### B.3 BASE RATE FOR MACHINE TRANSLATION

The base rate for machine translation is defined as the probability of observing a new word in the target at a particular time $t$ in training. We go through the unseen sentences in the corpus from the target language at time $t$ to compute the probability of sampling a sentence with at least one new word. Thus, the base rate at time $t$ in training is defined as:

$$P(\text{new in target at t}) = \frac{\text{\# of unseen sentences in target with new words}}{\text{\# of unseen sentences}}$$

### B.4 EXPERIMENT DETAILS FOR CLASSIFICATION

For Omniglot, a convolutional neural network was trained on 1623-way classification. The architecture consists of 3 convolutional layers (each consisting of $5 \times 5$ kernels and $64$ feature maps), a fully connected layer ($576 \times 128$) and a softmax classification layer. It was trained with a batch size of 16 using an Adam optimizer and a learning rate of 0.001. For Imagenet, a Resnet18 model (He et al., 2016) was trained on 1000-way classification with a batch size of 256, using an Adam optimizer and a learning rate of 0.001.

### B.5 CALCULATION OF $P(N|t)$ IN CLASSIFICATION

For classification, we calculate the score $P(N|t)$ for the model by adding the probabilities the model assigns to all the "new" (unseen) classes when iterating through the remaining corpus (similar to Equation 1). For the dataset, we compute $P(N|t)$ by sampling all unseen images in the corpus and compute the proportion from "new" classes given their ground truth labels.

