# OpenReview forum: "Mutual Exclusivity as a Challenge for Deep Neural Networks"
_ICLR.cc/2020/Conference — Reject_

### Official Review · AnonReviewer3 · 2019-10-22
**Official Blind Review #3**

**Rating:** 6

**Review:**

*** Increased to weak accept after discussion of merits of ME bias was improved in the paper ***

This paper investigates whether neural networks exhibit a ‘mutual exclusivity (ME) bias’, whereby novel inputs tend to be associated with previously unseen outputs, an inductive bias that is cited to be present in children when learning to associate new words and objects. Via three different sets of experiments, the authors conclude that, under standard training procedures, neural networks in fact display an anti-ME bias: when faced with a novel input, they tend to assign less probability mass to unobserved outputs than is justified by the incoming data. The authors go on to argue that explicitly designing neural networks to reason by mutual exclusivity could lead to faster and more flexible learning. While the experiments across several domains convincingly show the presence of an anti-ME bias in neural network training, I recommend that the paper be rejected because it falls short in demonstrating to what extent performance could be improved by correcting or reversing this bias. The authors pose an interesting hypothesis, but it would gain a lot of credibility if they could provide an empirical analysis of an algorithm that uses ME reasoning to improve learning in a realistic setting or if they could at least perform some quantitative analysis of the effects of ME bias on task performance for a toy example.

Comments / questions:
* The authors duly acknowledge that ME bias is not necessarily desirable in all circumstances, namely in tasks that feature many-to-one mappings, citing polysemy and synonymy in machine translation as examples. Without an empirical example in a realistic setting, it is hard to judge whether the benefits of introducing an ME bias for better classification of new inputs belonging to new classes can outweigh the negatives via a potential increase in misclassification of those belonging to old classes.
* For the synthetic dataset one-to-one mapping of one-hot vectors it is clear that an ME bias would incur an advantage for classification on a zero-shot basis, i.e. an increased an accuracy for the first time a new input is observed. Is ME bias considered to be useful here because (i) we care directly about improving this type of zero-shot classification, or is it (ii) because it's implicitly assumed that a better initial prediction will lead to faster learning on these examples?
    * If it is (i) we care about, then it would be useful to quantify the advantage gained either empirically or analytically. It’s not obvious to me how useful the ME bias would be when there is a large number of classes since the probability assigned to the true novel class with a perfect ME bias would be 1 / (# unseen classes). So at the beginning of training, while there are lots of new classes to benefit from the ME bias, the maximum advantage per example is small, and vice versa at the end of training - how do these conflicting effects balance each other out over the course of training and how do the benefits of ME bias scale with the number of classes? Presumably, whether the benefit scales well will depend on the loss function - in any case, this warrants some analysis/discussion especially given that there are no experiments to test it.
    * The authors say that “ME can be generalized from applying to 'novel versus familiar’ stimuli to instead handling ‘rare versus frequent’ stimuli” and they cite the fact that neural networks take longer to learn from rare stimuli, suggesting that ME could help for reason (ii) above. It’s not obvious to me that ME bias will provide a significant advantage in a one-shot or few-shot learning setting - i.e. how useful is having made a better initial guess (which is random amongst the unseen classes) after you gain some specific information about the target class? Perhaps the benefit is small relative to that of the one-shot or few-shot information gained - I think this needs to be quantified empirically.
* It would be interesting to see a more detailed analysis of the predictions of the image classifiers in section 4.2. When a new image is presented, how is the probability mass distributed across previously seen classes versus across unseen ones? From a strategic standpoint, in a scenario where it is difficult to determine whether a new image belongs to a new class or not, could it plausibly be more sensible for the network to make a strong prediction on its best guess out of previously seen classes (that it knows more about) rather than a scattered prediction on the unseen classes? Admittedly this is a difficult question to quantify, but my point is to question whether the anti-ME bias shown in Figure 6 is necessarily suboptimal, given the difficulty of classifying a new image as a new class or not.

Minor comments /questions not affecting review:
* Is the acronym ME pronounced like the word “me” or is it spelled out “M-E”? If the latter, then all cases of “a ME bias” should be corrected to “an ME bias”.
* Section 4.2 line 3: “sample the class [for the] from a power law distribution"

**Experience Assessment:**

I do not know much about this area.

**Review Assessment: Checking Correctness Of Derivations And Theory:**

I assessed the sensibility of the derivations and theory.

**Review Assessment: Checking Correctness Of Experiments:**

I assessed the sensibility of the experiments.

**Review Assessment: Thoroughness In Paper Reading:**

I read the paper thoroughly.

---

> ### Author Response · Authors · 2019-11-14
> **Thank you and please see above**
>
> We thank you for your constructive feedback. We reply to all reviews in a general response above.

---

### Official Review · AnonReviewer1 · 2019-10-24
**Official Blind Review #1**

**Rating:** 8

**Review:**

Summary:

This paper makes an observation that most of the neural network architectures do not learn the mutual exclusivity (ME) bias: if an object has one label, then it does not need another. Authors demonstrate this in both synthetic tasks and real-world tasks like object recognition and machine translation. Authors argue that ME bias could help the model to handle new classes and rare events better.

My comments:

I very much enjoyed reading this paper. I support accepting this paper. It highlights one of the missing inductive biases in ML and proposes it as a challenge. As the authors also agree, ME bias is missing not just in DNNs. It is the issue of MLE. It would be good to have some non-NN results too. I see this is a challenge for MLE than DNNs.

1. In figure-4 you mention that entropy regularizer helps to keep the initial ME score. Can you elaborate more about the way in which entropy regularizer is used with regular MLE training?
2. It is not very clear how is the base rate computed in Figure 5. I have a guess. But it is better to explain it in detail.
3. Section 4.2 need more clarity. For example, what do you mean by classifying the image as “new”? Is “new” a class name? Also, how is P(N|t) computed? Please explain.
4. Are the authors willing to release the code and data to reproduce the results?

Minor comments:


1. Page 3: second para, line 4: “our aim is to study”
2. Page 5: last line: estimate for -> estimated for
3. Section 4.2: 3rd line: “the class for the from”

=====================================================

After rebuttal: I have read the authors' response and  I stand by my decision.


**Experience Assessment:**

I have read many papers in this area.

**Review Assessment: Checking Correctness Of Derivations And Theory:**

N/A

**Review Assessment: Checking Correctness Of Experiments:**

I carefully checked the experiments.

**Review Assessment: Thoroughness In Paper Reading:**

I read the paper thoroughly.

---

> ### Author Response · Authors · 2019-11-14
> **Specific Response to R1**
>
> Thank you for your supportive review.
> We answer the specific queries below and have also added them to the revised version of the paper.
> 1.         We found that the entropy regularizer produces an ME score that stays constant across training, at the cost of the model being less confident about predictions made for seen classes. We added details regarding this condition to the manuscript.
> 2.         The base rate is the probability of observing a new word in the target at that particular point in training. We go through the remaining sentences in the corpus from the target compute the probability of sampling a sentence with at least one new word. Thus, the base rate at time t in training is defined as:
> $$P(\text{new in target at t}) = \frac{ \text{# of unseen sentences in target with new words}} {\text{# of unseen sentences}}$$
> 3.         In Section 4.2, we use “new” to refer to the set of all the unseen classes at a particular timepoint t. For the classifier, P(N|t) is calculated by adding the probabilities the model assigns to all the “new” classes when iterating through the remaining corpus (similar to Equation 1 in our paper). For the dataset, we compute P(N|t) by sampling all unseen images in the corpus and compute the proportion from “new” classes given their ground truth labels.
> 4.         We will release our code and data with the publication of the paper. Most of our experiments are easy to replicate as they use standard datasets, models, loss functions and optimizers. We sincerely hope that our challenge and these resources will stimulate progress in this area.
>
> Please also see above where we write a general response to all reviews.

---

### Official Review · AnonReviewer2 · 2019-10-25
**Official Blind Review #2**

**Rating:** 6

**Review:**

This paper targets at studying the mutual exclusive bias which existed in children learning, to help understand whether there exists similar bias in deep networks.

In general, the whole paper tries to tell a very interesting, and good story. The paper is very well organized and written. However, I have the following concerns.

1, the ME problem is quite similar to the concept ontology, e.g. , a “Dalmatian,” a “dog”, or a “mammal”.  So what’s the key difference? Hierarchical learners can avoid this problem.

2, I would say, the SOTA neural networks fundamentally are just representation learned, i.e., feature representation learning. The learned features could in principle be employed to construct advanced learners, .e.g., hierarchical Bayesian. It’s probably a bit unfair or misleading to claim neural networks suffering from ME bias.

3, In Sec. 3, the first and section paragraph, I can not quite understand how the ME bias theory guide the following synthetic experiments. Please give more explanations.

4, in Sec. 3.1, considering the small number of training instances, whether it is large enough to train the NN/ not overfitting issues?

5. Whether the ME bias mostly attributed to the one-hot representation? If one uses word2vec as the representation in NN, the ME bias will be solved. Actually, this is the standard practice in some learning tasks, e.g., zero-shot learning.

6, the experimental design of Sec. 4.2 is also a bit unfair. It seems to me that the tasks are organized as unbalanced instances of each classes, and asking the common NNs to learn this  task. Of course, this common NNs can not address it.

----
I read the rebuttal. the authors clarified and answered the questions. I would like to raise the score.

**Experience Assessment:**

I have read many papers in this area.

**Review Assessment: Checking Correctness Of Derivations And Theory:**

I assessed the sensibility of the derivations and theory.

**Review Assessment: Checking Correctness Of Experiments:**

I carefully checked the experiments.

**Review Assessment: Thoroughness In Paper Reading:**

I read the paper at least twice and used my best judgement in assessing the paper.

---

> ### Author Response · Authors · 2019-11-14
> **Thank you and please see general response above**
>
> Thanks for your feedback. We reply to all reviewers jointly in our comments above.

---

### Author Response · Authors · 2019-11-14
**General Response (1/2)**

We thank each of the reviewers for their thoughtful feedback on our work. We uploaded a revised paper that incorporates your suggestions, and we describe the changes in our response below. To echo R1, our paper “highlights one of the missing inductive biases in ML and proposes it as a challenge.” We demonstrate that popular deep neural network (DNNs) architectures do not show the mutual exclusivity (ME) bias that children use to help learn new words; in fact, they have an anti-ME bias that is completely backwards. We show how this anti-ME bias is poorly aligned with machine translation and object recognition tasks on common datasets, especially in more realistic lifelong learning settings. Our paper proposes a challenge to develop DNN architectures that can use the ME bias, like children, to support better zero-shot inferences and rapid learning.

As requested by R2, we clarify how the synthetic experiments (Section 3) implement the classic ME paradigm from Ellen Markman. As shown in Fig. 1 of our paper, children tend to select a novel object over a familiar object when asked to “Show me the dax” (Markman and Wachtel, 1988). Translating this into a synthetic experiment, input units denote words and output units denote objects. We ask the neural network to “Show me the dax” by activating the “dax” input unit and asking it to select amongst possible referents. The network produces a probability distribution a set of candidate referents, but it can make relative (two object) comparisons by isolating the scores of two candidates referents. To quantify the overall propensity toward ME, our “ME score” measures the relative probability of choosing any of the novel objects (as opposed to the familiar objects), which directly translates to higher scores on Markman’s forced choice task. We revised the task description in Section 3.1 to make these links clearer.

R2 offers suggestions for how to get models to exhibit the ME bias, and we appreciate your ideas here. We have been thorough in our explorations and have tried some of these, without success, which is why we see ME as such an interesting challenge! We have revised the paper to provide more details related to your suggestions, which we summarize here. R2 suggests that the anti-ME bias may arise from “the small number of training instances,” but we have verified that DNNs do not learn the ME regularity no matter how much data is presented, either in one-to-one synthetic mappings (Section 3) or on real datasets (Section 4). We clarify this in Section 3.1. The suggestion to use pre-trained embeddings (word2vec) is interesting but not pertinent to the ME paradigm which always tests novel words which are out of sample (“dax”, “zup,” “fep”, etc.), corresponding to novel concepts in a lifelong learning setting. Additionally, R2 suggests that adding hierarchy through a concept ontology or a hierarchical Bayesian learner might help induce the bias. This is an interesting idea that we can explore in future work, and we added it to the discussion section, but it’s not obvious to us that adding superordinate level categories like “animal” that have multiple referents would help map novel names to novel referents. Finally, with regards to the unbalanced classes in Section 4 (as mentioned by R2), an intelligent learner should be capable of modeling an open world that is inherently unbalanced, like children and adults do, with the possibility of encountering a new class at any point.

R3’s main critique is that it is hard to quantify the improvement ME would provide, in part because we do not “provide an empirical analysis of an algorithm that uses ME reasoning to improve learning.” Simply put, we can’t provide such an analysis because no such algorithm exists (yet)! This is why we are challenging the community to work in this direction, and we see our experiments as a demonstration that there will be applications on real tasks (Section 4). There is a clear misalignment between the inductive biases of standard neural nets and the statistical structure of common tasks especially when interpreted as lifelong learning. Recently, there have been some demonstrations of using ME to improve performance on specific tasks [Santoro et al. (2016), Lake, Linzen and Baroni (2019), Cohn-Gordon (2019), Lake (2019); see paper for bibliography], but none of these approaches represent a general solution. We see a general solution as an advance that will improve both zero-shot predictions and the speed of learning after just a few examples of a new input, by mitigating the strong bias to familiar responses.

Finally, we thank R1 for their positive feedback on our work. R1 asked for elaboration on some of the technical details of how some of the quantities are computed. We respond to your comment directly with answers to the questions and make corresponding updates to the main text.

---

> ### Author Response · Authors · 2019-11-14
> **General Response (2/2)**
>
> Our paper is unique for introducing a challenge that does not yet have a solution, but we see this as the best way forward for stimulating research in this important area. There is a wide gap between the centrality of ME in human language development and the strong anti-ME effects we found in standard DNNs, and MLE-based approaches more generally. We see our work as the beginning of a larger effort to integrate ME and inductive biases from cognitive science into modern AI models, and we thank you for considering our paper in your further discussions.

---

> ### Comment · AnonReviewer3 · 2019-11-14
> **Pros and cons of introducing an ME bias should be discussed in more depth**
>
> Thank you for your response. I appreciate that it is difficult to develop an algorithm that corrects the ME bias since it’s not trivial to decide whether an input is from a new class or not, but given that such an algorithm is not put forth, I don’t think enough justification is provided that directly inducing this bias will help performance. It seems to me that there are circumstances where it’s not clear that inducing an ME bias is a good idea and, while I understand that in some cases it may be useful, it may be harmful to it an explicit objective - for this reason, I think there should be more of a discussion of when it is useful, when it is not, and how it can go wrong.
> In the 5th bullet point of my review, I tried to make the point that it’s not obvious that an ME bias is strategically smart for a neural network that is uncertain whether a new input belongs to a new class or not, especially when the total number of classes is large. For example, if the network has seen lots of black bears and gets a picture of a brown bear (a class its never seen before), it might be a better bet to make a strong guess that it’s another black bear rather than make a scattered prediction across the (potentially numerous) previously unseen classes. For a similar reason, as mentioned in bullet point 3, an ME bias early on in training might not be very useful since there are so many unseen classes. As mentioned in my review, it would be interesting to see how the network distributes the class probabilities of new inputs for this reason. In general, there seems to be a tradeoff between better classification of new classes and potential misclassification of old classes that would be introduced by inducing an ME bias that could be affected by factors such as the similarity of groups of classes, number of classes, stage of training and the loss function. While there is a line in section 4.1 mentioning that the “answer to whether or not ME holds is not absolute”, citing polysemy and synonymy as examples, I think it’s important that the pros and cons are fleshed out more, given that the paper advocates for introduction of ME bias into training.

---

> > ### Author Response · Authors · 2019-11-15
> > **A discussion on the utility of ME**
> >
> > Thanks for continuing the discussion of our work. Your point is well received and we agree that the ME bias is not appropriate for every use case. Moreover, even if it is optimal given the inherent tradeoffs in any inductive bias, it’s bias that is going to get some cases right and other cases wrong. As you rightly point out, the issue is complex and warrants more nuance in the discussion, and in response we uploaded another revision that expands the discussion in this way (see Section 5). We summarize the main points here too. As we say, the ME bias may not be helpful in every case, but it’s equally clear that the status quo is sub-optimal: models shouldn’t have a strong anti-ME bias regardless of what the task and dataset demands. So how do we proceed? An ideal model would decide for itself how strongly to use ME based on the task demands. For instance, in our synthetic example, an ideal learner would discover the one-to-one correspondence and use this as a meta-strategy (through a perfect ME bias). If the dataset has more many-to-one correspondences, it would adopt another meta-strategy. This meta-strategy could even change depending on the stage of learning. There are promising results in this direction: Santoro et al. (2016) and Lake (2019) [see references in paper] do not build a ME strategy into the model, but rather observe one as an emergent consequence of meta learning. We see potential in this direction but there are other means of tackling the challenge too. We hope that by introducing this challenge, our paper will stimulate debate and ultimately progress in addressing it. Our discussion is now more appropriately nuanced, and we thank you again for your feedback on this point.

---

> > > ### Comment · AnonReviewer3 · 2019-11-15
> > > **Improved discussion and qualification of merits of ME bias - increasing to weak accept**
> > >
> > > Though I still think the paper would ideally benefit from some more in-depth and specific analysis of when ME bias is desirable (and I hope to see in future work!), I think that it is strengthened by the changes made to the discussion described above and I am willing to increase my score to weak accept.

---

### Decision · Program_Chairs · 2019-12-19

**Decision:**

Reject

**Comment:**

This paper presents an understudied bias known to exist in the learning patterns of children, but not present in trained NN models.  This bias is the mutual exclusivity bias: if the child already knows the word for an object, they can recognize that the object is likely not the referent when a new word is introduced.  So that is, the names of objects are mutually exclusive.

The authors and reviewers had a healthy discussion. In particular, Reviewer 3 would have liked to have seen a new algorithm or model proposed, as well as an analysis of when ME would help or hurt.  I hope these ideas can be incorporated into a future submission of this paper.